# Dpp from the anterior stripe of cells is crucial for the growth of the *Drosophila* wing disc

**Shinya Matsuda\*, Markus Affolter\***

Biozentrum der Universität Basel, Basel, Switzerland

**Abstract** The Dpp morphogen gradient derived from the anterior stripe of cells is thought to control growth and patterning of the *Drosophila* wing disc. However, the spatial-temporal requirement of *dpp* for growth and patterning remained largely unknown. Recently, two studies re-addressed this question. By generating a conditional null allele, one study proposed that the *dpp* stripe is critical for patterning but not for growth (Akiyama and Gibson, 2015). In contrast, using a membrane-anchored nanobody to trap Dpp, the other study proposed that Dpp dispersal from the stripe is required for patterning and also for medial wing disc growth, at least in the posterior compartment (Harmansa et al., 2015). Thus, growth control by the Dpp morphogen gradient remains under debate. Here, by removing *dpp* from the stripe at different time points, we show that the *dpp* stripe source is indeed required for wing disc growth, also during third instar larval stages.

**\*For correspondence:** shinya.
matsuda@unibas.ch (SM); markus.
affolter@unibas.ch (MA)

**Competing interests:** The authors declare that no competing interests exist.

## Introduction

Morphogens are thought to disperse and form concentration gradients to control tissue patterning and growth (*Rogers and Schier, 2011*). The *Drosophila* wing imaginal disc has served as an excellent model to study how morphogens control patterning and growth. It has been shown that *decapentaplegic* (*dpp*), a homologue of vertebrate bone morphogenetic protein 2/4 (BMP2/4), is expressed in a stripe of cells in the anterior compartment along the anterior-posterior compartmental boundary of the wing imaginal disc. From this source, Dpp protein is thought to spread and form a concentration gradient to control patterning and growth of the wing imaginal disc (*Lecuit et al., 1996*; *Nellen et al., 1996*; *Matsuda et al., 2016*; *Affolter and Basler, 2007*; *Restrepo et al., 2014*). However, the precise spatial-temporal requirement of *dpp* remained elusive, mainly because it was not possible to generate an inducible null allele of *dpp* due to the haploinsufficiency of the locus and the lack of appropriate methods. Recently, two papers addressed the role of the *dpp* source at the compartment boundary using independent strategies. Akiyama and Gibson used CRISPR-Cas9-mediated genome engineering techniques to insert a FRT (Flippase Recognition Target) cassette into the *dpp* locus, and successfully generated a conditional null allele of *dpp* using the expression of Flippase (FLP) in a spatial-temporal controlled manner (*Akiyama and Gibson, 2015*). By genetically removing *dpp* from the anterior stripe, they showed that the Dpp morphogen gradient derived from this stripe is indeed critical for patterning. However, and rather surprisingly, they also found that *dpp* from the stripe is largely dispensable for wing disc growth during the third instar larval stage. Instead, growth of the wing disc was compromised by genetically removing *dpp* from the entire anterior compartment. Based on the constant requirement of *dpp* derived from the anterior compartment for growth of the wing disc, Akiyama and Gibson proposed that a not-yet identified anterior *dpp* source outside the stripe of cells is required for wing disc growth (*Akiyama and Gibson, 2015*).

**eLife digest** From the wings of a butterfly to the fingers of a human hand, living tissues often have complex and intricate patterns. Developmental biologists have long been fascinated by the signals – called morphogens – that guide how these kinds of pattern develop. Morphogens are substances that are produced by groups of cells and spread to the rest of the tissue to form a gradient. Depending on where they sit along this gradient, cells in the tissue activate different sets of genes, and the resulting pattern of gene activity ultimately defines the position of the different parts of the tissue.

Decades worth of studies into how limbs develop in animals from mice to fruit flies have revealed common principles of morphogen gradients that regulate the development of tissue patterns. Morphogens have been shown to help regulate the growth of tissues in a number of different animals as well. However, how the morphogens regulate tissue size and what role their gradients play in this process remain topics of intense debate in the field of developmental biology.

In the developing wing of a fruit fly, a morphogen called Dpp is expressed in a thin stripe located in the center and spreads to the rest of the tissue to form a gradient. Matsuda and Affolter have now characterized where and when the Dpp morphogen must be produced to regulate both the final size of the fly's wing and the number of cells the wing eventually contains. The experiments involved preventing the production of Dpp in the developing wing in specific cells and at specific stages of development. This approach confirmed that Dpp must be produced in the central stripe for the wing to grow. Bosch, Ziukaite, Alexandre et al. and, independently, Barrio and Milán report the same findings in two related studies, and also conclude that the gradient of Dpp throughout the wing is not required for growth.

Further work will be needed to explain how the Dpp signal regulates the growth of the wing. The answer to this question will contribute to a better understanding of the role of morphogens in regulating the size of human organs and how a failure to do so might cause developmental disorders.

Harmansa et al. used a membrane-anchored anti-GFP nanobody (morphotrap) to trap GFP-Dpp and manipulate GFP-Dpp dispersal (*Harmansa et al., 2015*). Since an endogenously tagged *GFP-Dpp* strain was not available, *dpp* disc mutants were rescued by expressing GFP-Dpp in the stripe (*Entchev et al., 2000*; *Teleman and Cohen, 2000*) and morphotrap was concomitantly expressed in the stripe in order to trap GFP-Dpp and block its dispersal. In this setup, the authors confirmed that *dpp* is required for wing disc patterning, and also found that Dpp morphogen dispersal from the stripe of cells is required for medial but not for lateral wing disc growth in the posterior compartment (the region they analyzed). However, since these experiments were done under rescue conditions, other sources of *dpp* important for growth in wild type individuals could have been missed. Thus, while both studies confirmed a role of *dpp* on wing disc patterning, these studies propose different scenarios for the spatial requirement of *dpp* on wing disc growth, and it remains debated whether the Dpp morphogen gradient derived from the anterior stripe of cells is required for wing disc growth (*Vincent et al., 2016*; *Strzyz, 2016*). In this study, we first show that the *dpp*-Gal4 driver line used to remove *dpp* from the anterior stripe in the previous study (*Akiyama and Gibson, 2015*) does not faithfully reflect the endogenous *dpp* expression pattern during third instar larval stages. We therefore genetically removed *dpp* at different time points using a different Gal4 line (*ptc*-Gal4 line), which covers the anterior stripe of cells from the early larval stages onward. Using this setup, we demonstrate that *dpp* from the stripe of cells is indeed critical for growth of the wing disc, even during third instar larval stages. Furthermore, this result indicates that an anterior *dpp* source outside the stripe of cells, even if it would exist, would not be sufficient to drive growth of the wing disc.

## Results and discussion

Akiyama and Gibson used a *dpp*-Gal4 driver line to genetically remove *dpp* from the anterior stripe. Based on the results they obtained, they proposed that the *dpp* stripe is not required for wing disc growth (*Akiyama and Gibson, 2015*). Although it appears straightforward to use *dpp*-Gal4 to

remove *dpp*, this setup has intrinsic problems in removing *dpp* from the early onset of its expression. Since this setup requires the *dpp* disc enhancer to be activated, endogenous *dpp* is initially expressed before it can be removed by FLP/FRT recombination. Furthermore, since it takes about 18–24 hr to remove the engineered cassette in the *dpp* locus from the majority of cells in the anterior stripe and the wing disc grows dramatically during this time (*Akiyama and Gibson, 2015*) (*Figure 1a*), this delay may fail to reveal a potential early function of *dpp* on growth. In addition to this intrinsic problem, we found that *dpp*-Gal4 expression does not faithfully reflect the spatial-temporal endogenous *dpp* expression pattern until relatively late third instar stages (*Figure 1a*). Although *dpp* has been shown to be expressed in the entire anterior stripe of cells from the early third instar larval stages (*Akiyama and Gibson, 2015*), NLS-mCherry expressed under the control of *dpp*-Gal4 marked only the dorsal stripe of cells at the beginning of third instar larval stages (60 hr after egg laying (AEL) at 26°C) (*Figure 1a*). *dpp*-Gal4 was expressed in the entire stripe of cells only in relatively late third instar stages (*Figure 1a*). Most probably, the fragment of the *dpp* disc enhancer used to drive Gal4 does not cover all the *cis*-regulatory regions important for proper stripe expression.

Therefore, to reinvestigate the role of the *dpp* stripe on growth, we decided to remove *dpp* by FLP/FRT recombination using a different Gal4 line expressed in this anterior stripe of cells. Since *ptc*, a Hedgehog target gene, is expressed in a stripe of cells similar to *dpp*, we first analyzed the spatial-temporal expression pattern of *ptc*-Gal4. We found that NLS-mCherry expressed under the control of *ptc*-Gal4 marked the entire stripe of cells as early as late second instar (50 hr AEL at 26°C) and that stripe expression continued throughout third instar larval stages (*Figure 1b*). We then compared the expression pattern of these two Gal4 lines with endogenous Dpp expression using a Dpp antibody that recognizes the prodomain of Dpp (*Akiyama and Gibson, 2015*) (*Figure 1c–f*). Consistent with the above observations, *dpp*-Gal4 expression did not cover the ventral Dpp stripe in the early third instar larval stages and covered the entire stripe only in late third instar larval stages (*Figure 1c,d*). In contrast, *ptc*-Gal4 expression continuously covered the Dpp stripe from the early third instar larval stages onward (*Figure 1e,f*). To ascertain the validity of using NLS-mCherry to mark *ptc*-Gal4 or *dpp*-Gal4 expression, which might be somewhat compromised due to the lengthy maturation time for the mCherry protein, we marked Gal4 expression using α-mCherry antibody and found that fluorescent signal and antibody signal overlap well (*Figure 1—figure supplement 1*). Taken together, these results show that spatial-temporal *ptc*-Gal4 expression reflects endogenous *dpp* expression pattern in the wing disc more precisely than *dpp*-Gal4.

We then removed *dpp* using *dpp*-Gal4 or *ptc*-Gal4 and compared their effects on wing disc growth (CRISPR-Cas9-modified flies (*dpp*^FO^) were generously provided by Akiyama and Gibson). When *dpp* was removed using *dpp*-Gal4, pMad was not detectable in the wing pouch but the wing disc grew normally, as reported in Akiyama and Gibson (*Figure 2a,b*) (*Akiyama and Gibson, 2015*). In sharp contrast, when *dpp* was removed using *ptc*-Gal4, the pMad signal was also lost from the wing pouch (except for the future alula region), and wing disc growth was severely affected (*Figure 2c*). Wing pouches were very small and often hardly visible, as shown by the lack of internal ring expression of Wg (*Figure 2c*). We confirmed that Dpp expression was lost from the stripe of anterior cells but remained detectable in the future alular region (*Figure 2d*), consistent with the pMad signal in this region (*Figure 2c*). Accordingly, the growth repressor Brk, normally repressed by pMad, was uniformly expressed in the wing pouch (*Figure 2e*).

The above results show that wing disc growth is severely affected by removing *dpp* using *ptc*-gal4. To test where FLP/FRT recombination was driven by *ptc*-Gal4, we then marked cell lineages of *ptc*-Gal4 (*Figure 2f–i*). Since there is no lineage separation within the anterior compartment, the temporal Gal4 expression pattern does not necessarily reflect the actual domain, in which FLP/FRT-mediated recombination occurred. For example, cells that leave the stripe in the early stages might not express NLS-mCherry anymore at later stages, but could have excised *dpp* when they were located in the stripe in earlier stages. Alternatively, Gal4 may be transiently expressed outside the stripe in the early stages and be sufficient to excise *dpp* there. We found that when one copy of *dpp* was removed using *ptc*-Gal4 (control), the entire anterior wing pouch was marked in some wing discs, and only a stripe of anterior cells was marked in other wing discs (*Figure 2f,g*). Similar cell lineages were also observed when two copies of *dpp* were removed using *ptc*-Gal4 (*Figure 2h,i*). The variations in these *ptc*-Gal4 lineages observed may be due to slight differences in Gal4 or FLP expression or a degree of randomness in the excision events in each wing disc. Importantly, the wing discs where the *ptc*-Gal4 lineages

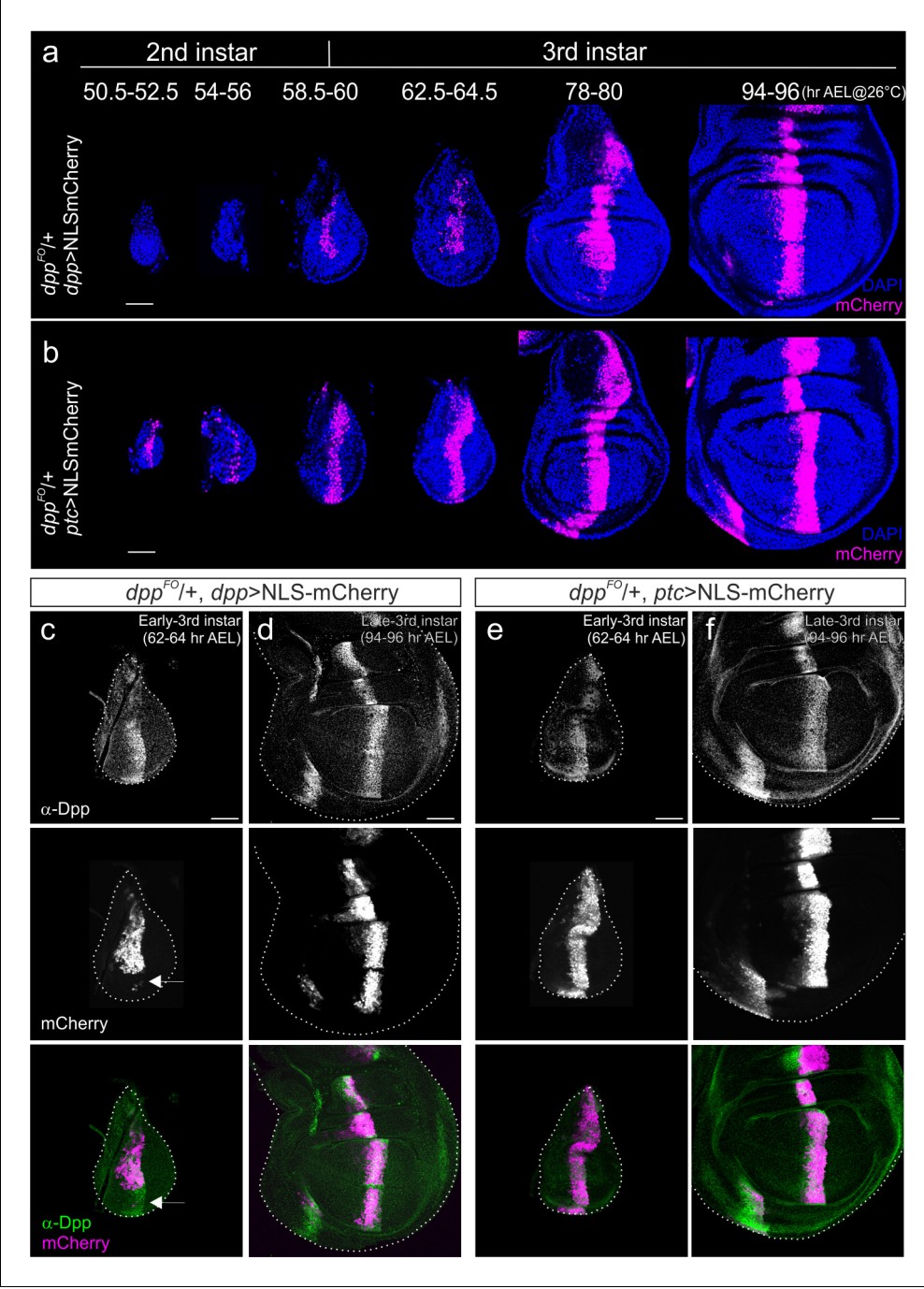

**Figure 1.** Comparison of *dpp*-Gal4 and *ptc*-Gal4 expression pattern with Dpp expression in the *Drosophila* wing disc. (a) Temporal expression pattern of *dpp*-Gal4 ($dpp^{FO}$/+; *dpp*-Gal4/UAS-NLS-mCherry) (b) temporal expression pattern of *ptc*-Gal4 ($dpp^{FO}$, *ptc*-Gal4/+; UAS-NLS-mCherry). Single confocal images except 50.5–52.5 hr AEL by maximum intensity projection. (c, d) Comparison of anti-Dpp staining and *dpp*-Gal4 expression (NLS-mCherry) in the early (c) and late (d) third instar wing disc of a $dpp^{FO}$/+; *dpp*-Gal4/UAS-NLS-mCherry larva. (e, f) Comparison of anti-Dpp staining and *ptc*-Gal4 expression (NLS-mCherry) in the early (e) and late (f) third instar wing disc of a $dpp^{FO}$/+; *ptc*-Gal4/UAS-NLS-mCherry larva. Average intensity projection from 5 sequential confocal images. Scale bars 50 μm. Anterior is to the left in all figures.

The following figure supplement is available for figure 1:

**Figure supplement 1.** Comparison of mCherry fluorescent signal and anti-mCherry staining.

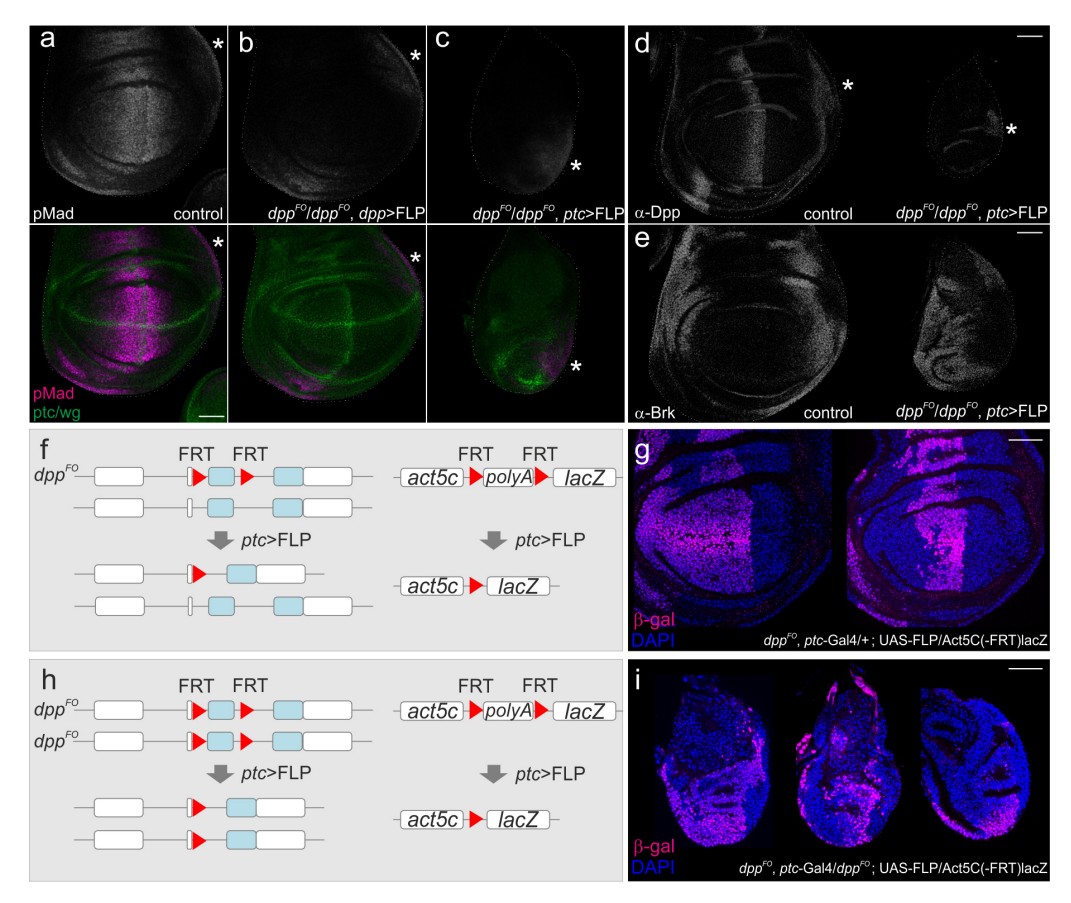

**Figure 2.** Defects in wing disc growth by removing *dpp* using *ptc*-Gal4. (a–c) Anti-pMad and anti-ptc/wg staining in a *dpp^FO*/+; UAS-FLP/+ (control) late third instar wing disc (a), in a *dpp^FO*/*dpp^FO*; *dpp*-Gal4/UAS-FLP late third instar wing disc (b), and in a *dpp^FO*, *ptc*-Gal4/*dpp^FO*; UAS-FLP/+ late third instar wing disc (c). (d) anti-Dpp staining in a *dpp^FO*, *ptc*-Gal4/+; UAS-FLP/+ (control) late third instar wing disc (left), and in a *dpp^FO*, *ptc*-Gal4/*dpp^FO*; UAS-FLP/+ late third instar wing disc (right). (e) anti-Brk staining in a *dpp^FO*, *ptc*-Gal4/+; UAS-FLP/+ (control) late third instar wing disc (left), and in a *dpp^FO*, *ptc*-Gal4/*dpp^FO*; UAS-FLP/+ late third instar wing disc (right). (*) marks the future alula region. (a–e) Average intensity projection from 5 sequential confocal images. (f, h) an experimental setup to test the efficiency of FLP/FRT mediated recombination. (g) anti-*β*-gal staining in a *dpp^FO*, *ptc*-Ga4/+; UAS-FLP/act5C(-FRT)lacZ late third instar wing disc (control). (i) anti *β*-gal staining in a *dpp^FO*, *ptc*-Gal4/*dpp^FO*;UAS-FLP/act5C(-FRT)lacZ late third instar wing disc. (g, i) A single confocal image. Scale bars 50 μm. Anterior is to the left in all figures.

were restricted to a stripe of cells showed drastic growth defects (*Figure 2i*), raising a possibility that *dpp* derived from the anterior stripe is critical for wing disc growth.

However, since *ptc*-Gal4 is expressed earlier than *dpp*-Gal4 (*Figure 1*), the severe growth defects by *ptc*-Gal4 may simply reflect an early role of *dpp* stripe on wing disc growth. Furthermore, since different FRT cassettes can have different sensitivities to FLPase, the FRT cassette in *dpp* locus and the FRT cassette to follow the *ptc*-Gal4 lineage could be excised in different regions. For example, the FRT cassette in the *dpp* locus may be excised outside the anterior stripe, although the excisions of the FRT cassette to follow the *ptc*-Gal4 lineages are restricted to the anterior stripe. Thus, the growth defect by *ptc*-Gal4 may be due to elimination of *dpp* in the early stages, and/or elimination of potential *dpp* source outside the anterior stripe that may drives wing disc growth.

To investigate the temporal requirement of the *dpp* stripe for wing disc growth, we therefore used the *Gal80ts* system (*Figure 3a*). Gal80ts represses Gal4 activity at 17°C and can be inactivated at 29°C. Thus, *ptc*-Gal4 can be conditionally activated upon a temperature shift. Indeed, we found that at the permissive temperature *dpp^FO*, *ptc*-Gal4 / *dpp^FO*; *tub*-Gal80ts / UAS-FLP adult wing had no obvious wing phenotype, suggesting that Gal80ts effectively suppresses *ptc*-Gal4 activity at 17°C (*Figure 3—figure supplement 1*).

We first marked lineages of *ptc*-Gal4 when FLP/FRT-mediated recombination was temporally activated at the beginning of the second or third instar larval stage using Gal80ts. We found that *ptc*-Gal4 lineages were strictly restricted to the anterior stripe in control and mutant wing disc (*Figure 3b,d*). Thus, the more random lineages shown in *Figure 2g,i* appear to be derived from early expression of the *ptc*-Gal4 driver. To directly monitor where *dpp* is removed by conditionally activating *ptc*-Gal4, we utilized the *dpp^{FO-GFP}* allele originally generated as an intermediate allele to generate the final *dpp^{FO}* allele (*Akiyama and Gibson, 2015*). *dpp^{FO-GFP}* contains the same FRT cassette as *dpp^{FO}* and a ubiquitously expressed GFP construct (ubi-GFP) within the cassette (*Figure 3—figure supplement 2*). Thus, the regions in which *dpp* is excised will lack the GFP signal. Using this setup, we found that FLP/FRT-mediated excision in the *dpp* locus was indeed restricted to the anterior stripe from second and from third instar larval stages although the excision varied within the stripe (21/21 and 14/14 wing discs, respectively) (*Figure 3—figure supplement 2*). This result also strongly suggests that the temporal lineages of *ptc*-Gal4 indeed reflect the actual region where *dpp* is removed in this setup (*Figure 3b,d*).

By removing *dpp* during either second or third instar larval stages, we found that wing disc growth was severely affected in both cases, as measured in the late third instar larval stages (53 hr or 43 hr later after temperature shift, respectively) (*Figure 3b'–e'*). The majority of Dpp was eliminated around 20 ~ 24 hr after temperature shift as also reported by Akiyama and Gibson (*Akiyama and Gibson, 2015*) (*Figure 3—figure supplement 3*) and wing disc appeared to grow at least until 24 hr after temperature shift. Thus, the growth defects resulting from temperature shifting at the beginning of third instar larval stages likely reflect the effects seen from the absence of *dpp* around mid-third instar larval stages. These results show that the *dpp* stripe is required for wing disc growth during second and even third instar larval stages.

Since we used the same *dpp* allele (*dpp^{FO}*) and UAS-FLP line as Akiyama and Gibson, the different growth defects observed should be due to the differences between the *dpp*-Gal4 and *ptc*-Gal4 driver lines. We showed that while *ptc*-Gal4 is constantly expressed at the anterior stripe during third instar larval stages, *dpp*-Gal4 is initially expressed only at the dorsal stripe and only later, during third instar larval stages, in the entire stripe (*Figure 1*). Thus, the differences in the spatial-temporal expression may be responsible for the different phenotype observed when using *dpp*-Gal4 or *ptc*-Gal4. Interestingly, although both *dpp*-Gal4 and *ptc*-Gal4 are expressed in the dorsal stripe in the early third instar larval stages, the dorsal wing disc pouch still grew using *dpp*-Gal4 with minor growth defects (*Figure 2b*) (*Akiyama and Gibson, 2015*). To investigate why the dorsal compartment still grows when using *dpp*-Gal4, we analyzed Brk expression, the critical growth repressor repressed by Dpp signaling. (*Figure 3—figure supplement 4*). We found that at the mid third instar larval stage (80 hrAEL at 26°C), Brk was repressed in the ventral compartment where *dpp* was still expressed but was slightly upregulated in the dorsal compartment where the majority of *dpp* was removed (*Figure 3—figure supplement 4*). Interestingly, Brk upregulation in the dorsal compartment was not uniform but graded; lower in ventral and higher in dorsal regions within the dorsal compartment (*Figure 3—figure supplement 4*). At the late third instar larval stage, the majority of *dpp* was eliminated and Brk was upregulated in both dorsal and ventral compartment but again, Brk was not uniformly upregulated (*Figure 3—figure supplement 4*). Consistent with our findings, Omb has been shown to be weakly expressed in this setup (*Akiyama and Gibson, 2015*). These results suggest that the Dpp signal is not completely removed from the wing imaginal disc when using *dpp*-Gal4. The graded Brk expression in the dorsal compartment is consistent with weak Dpp signal derived from the ventral *dpp* stripe, and this lasting signal can explain the sustained growth there with minor growth defects. In contrast, when *dpp* was removed by *ptc*-Gal4 during the third instar larval stages, majority of *dpp* was removed from the entire anterior stripe (*Figure 3—figure supplement 3*) and wing disc growth was severely affected (*Figure 3b–d*).

Together, these results suggest that the critical role of the *dpp* stripe on wing disc growth was missed by Akiyama and Gibson due to imprecise spatial removal of *dpp* when using *dpp*-Gal4 during third instar larval stages (*Akiyama and Gibson, 2015*). Based on the constant requirement of *dpp* from the anterior compartment during third instar larval stages for proper wing disc growth, Akiyama and Gibson further proposed that a potential anterior *dpp* source outside the stripe was critical for wing disc growth (*Akiyama and Gibson, 2015*). However, our data show that wing disc growth was severely affected when *dpp* was removed only from the anterior stripe during third instar larval stages. Thus, the strong growth defects resulting from removing *dpp* from the entire

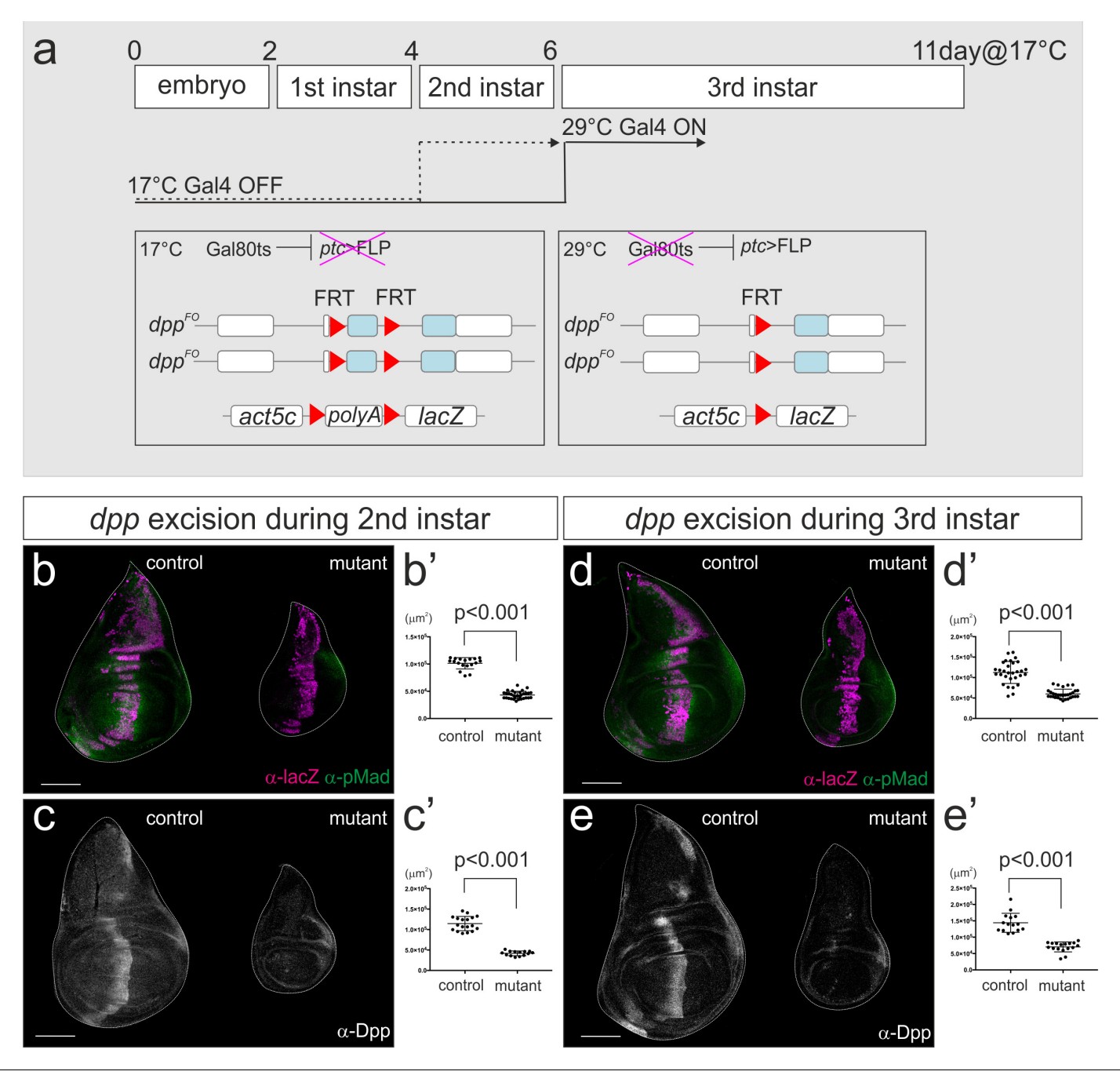

**Figure 3.** *dpp* stripe is required for wing disc growth during second and third instar larval stages. (**a**) A scheme to genetically remove *dpp* from the second or third instar larval stages using Gal80ts system. At 17°C, Gal4 activity is blocked by Gal80ts. At 29°C, Gal80ts is inactivated and Gal4 starts to induce FLP expression in the anterior stripe of cells. After embryo collection for 2–4 hr at room temperature, the embryos were incubated at 17°C until temperature shift. Temperature was shifted after 4 days (second instar) or 6 days (third instar) at 17°C, and late third instar wing discs were dissected after 53 hr or 43 hr later respectively. (**b–e**) Removal of *dpp* using *ptc*-Gal4 during second instar larval stages (**b, c**) or third instar larval stages (**d, e**). (**b, d**) anti-pMad staining and anti-*β*-gal staining (lineage tracing) in a control male wing disc (*dpp*^FO^, *ptc*-Gal4/CyO; *tub*-Gal80ts/UAS-FLP, act5C(-FRT)lacZ) (left), and a male wing disc removing *dpp* during the specified time point (*dpp*^FO^, *ptc*-Gal4/*dpp*^FO^; *tub*-Gal80ts/UAS-FLP, act5C(-FRT)lacZ) (right). (**c, e**) anti-Dpp staining in a control male wing disc (*dpp*^FO^, *ptc*-Gal4/+; *tub*-Gal80ts/ +) (left), and a male wing disc removing *dpp* during the specified time point (*dpp*^FO^, *ptc*-Gal4/*dpp*^FO^; *tub*-Gal80ts/UAS-FLP) (right). (**b'–e'**) Quantification of the wing disc size of (**b–e**). Mean ± s.d. p<0.001 by two sided Student's *t*-test. Scale bars 100 μm.

The following source data and figure supplements are available for figure 3:

*Figure 3 continued on next page*

*Figure 3 continued*

**Source data 1.** Quantification of wing disc size for *Figure 3b–e*.
**Figure supplement 1.** A control experiment under permissive temperature (17°C) for *Figure 3*.
**Figure supplement 2.** Visualization of the regions where *dpp* is removed using *ptc*-Gal4 in the wing imaginal disc.
**Figure supplement 3.** Temporal resolution of *dpp* removal using *ptc*-Gal4.
**Figure supplement 4.** Temporal changes in Dpp and Brk expression by removing *dpp* using *dpp*-Gal4.

anterior compartment are most likely due to excision of *dpp* from the anterior stripe and not due to the excision of the potential *dpp* outside the stripe.

In conclusion, our results establish that the anterior *dpp* stripe is critical for growth as well as patterning of the wing imaginal disc. Given the slow process of removing *dpp* by FLP/FRT mediated recombination (about 20–24 hr) compared to wing disc growth, it remains an open question whether the requirement of the *dpp* stripe on wing disc growth changes over time. It would be important to acutely manipulate the endogenous morphogen gradient at the protein level to address the precise temporal requirement of the *dpp* stripe on wing disc growth (*Matsuda et al., 2016*; *Bieli et al., 2016*).

## Materials and methods

### Fly stocks
Flies were kept in standard fly vials (containing polenta and yeast) in a 26°C incubator. The following fly lines were used: $dpp^{FO}$, $dpp^{FO-GFP}$, *dpp*-Gal4, and UAS-FLP (Matthew Gibson), UAS-NLS-mCherry (*Caussinus et al., 2008*), *ptc*-Gal4 (w*; P{GawB}ptc559.1), P{act5C(FRT.polyA)lacZ.nls1}3, ry506, *tub*-Gal80ts (Bloomington stock center).

### Immunostainings and antibodies
Protocol was described previously (*Harmansa et al., 2015*). Each fly cross was set together with control and >10 wing imaginal discs from each genotype were processed in parallel. If the genotype could be distinguished, experimental and control samples were processed in the same tube. A representative wing disc was shown for all the experiments. Following primary antibodies were used; anti-Dpp (1:100; Matthew Gibson), anti-phospho-Smad1/5 (1:200; Cell Signaling, 9516S), anti-Brk (1:1000; Gines Morata), anti-Wg (1:120; DSHB, University of Iowa), anti-Ptc (1:40; DSHB, University of Iowa), anti-$\beta$-Galactosidase (1:1000; Promega Z378A), anti-mCherry (1:5000; Nigg lab, University of Basel). All the primary antibodies except anti-Dpp antibody were diluted in 5% normal goat serum (NGS) (Sigma) in PBT (0.03% Triton X-100/PBS). Anti-Dpp antibody was diluted in 5% NGS in Can Get Signal Immunostain Solution B (TOYOBO). All secondary antibodies from the AlexaFluor series were used at 1:500 dilutions. Wing discs were mounted in Vectashield (H-1000, Vector Laboratories). Images of wing discs were obtained using a Leica TCS SP5 confocal microscope (section thickness 1 μm).

## Acknowledgements
We thank Akiyama and Gibson for flies; the Biozentrum Imaging Core Facility for maintenance of microscopes and support; the Developmental Studies Hybridoma Bank at The University of Iowa for antibodies; Dimitri 'Chin-Chin' Bieli and Gustavo Aguilar for discussion, sharing reagents, and comments on the manuscript. We would like to thank Bernadette Bruno, Gina Evora and Karin Mauro for constant and reliable supply with world's best fly food. SM is supported by a JSPS Postdoctoral Fellowship for Research Abroad. The work in the laboratory was supported by grants from Cantons Basel-Stadt and Basel-Land and from the SNSF (MA).

## Additional information

### Funding

| Funder | Grant reference number | Author |
|---|---|---|
| Basel-Stadt | | Markus Affolter |
| Basel-Land | | Markus Affolter |
| Japan Society for the Promotion of Science | Postdoctoral Fellowship for Research Abroad | Shinya Matsuda |

The funders had no role in study design, data collection and interpretation, or the decision to submit the work for publication.

### Author contributions

SM, Conceptualization, Formal analysis, Validation, Investigation, Visualization, Methodology, Writing—original draft, Writing—review and editing; MA, Supervision, Funding acquisition, Writing—review and editing

### Author ORCIDs

Shinya Matsuda, http://orcid.org/0000-0002-7541-7914
Markus Affolter, http://orcid.org/0000-0002-5171-0016

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
