## [Decision Letter]

Thank you for submitting your article "Dpp from the anterior stripe of cells is crucial for the growth of the *Drosophila* wing disc" for consideration by *eLife*. Your article has been reviewed by three peer reviewers, and the evaluation has been overseen by a Reviewing Editor and K VijayRaghavan as the Senior Editor. The reviewers have opted to remain anonymous.

The reviewers have discussed the reviews with one another and the Reviewing Editor has drafted this decision to help you prepare a revised submission. We hope you will be able to submit the revised version within two months.

Summary:

A contentious item that continues to raise interest concerns the relationship between the gradient of the BMP4-like signaling protein Dpp produced in *Drosophila* wing discs and cell proliferation in the disc. Dpp is both necessary and sufficient for disc growth, and the problem basically boils down to why regions with different levels of Dpp and BMP signaling do not cause different amounts of growth. Evidence and arguments on this point remain of high current interest.

The present manuscript is a partial rebuttal to the 2015 Nature paper from Akiyama and Gibson, which used various types of dpp loss-of-function clones to argue that the BMP Dpp produced by the stripe of cells anterior to the A/P boundary in wing discs was not necessary for the growth of the disc. This argued that models based on reading a gradient of Dpp were likely wrong, and that levels could be greatly reduced without greatly affecting growth, and thus models based on a temporal gradient of increasing BMP signaling were probably wrong as well. Nonetheless, that study showed that the Dpp produced by the entire anterior compartment was necessary for growth, at least up until 36 hours before wandering third instar, presumably this was supplied by low-level Dpp produced outside the normal stripe of high level Dpp expression.

The present manuscript argues that stripe Dpp is necessary for growth, using the same conditional *dpp* allele used by Akiyama, but a different (*ptc*) Gal4 driver that the authors show covers more of the stripe at earlier stages, especially in the ventral pouch.

Essential revisions:

Reviewer 1:

1) Since Akiyama already showed that removal from the entire anterior compartment reduces growth, the claim in the present study rests entirely on whether *ptc*-gal4 drives the excision of the conditional *dpp* allele in the stripe, or whether it drives excision more widely in the disc. This is a real worry, because the endogenous *ptc* gene is expressed at low levels throughout the anterior compartment. And as the authors (and others previously) show, *ptc*-gal4 can drive excision of a G-TRACE maker throughout the anterior as well.

Unfortunately, there is no direct way of telling where the *dpp* allele has been recombined. The conditional allele contains no marker of excision, and loss of dpp itself cannot be detected except where dpp expression is very strong, as Akiyama did.

The authors argue that there are some discs where *ptc*-gal4-driven G-TRACE is not excised throughout the anterior, and that therefore these discs must also be those where the conditional dpp allele is not recombined throughout the anterior. However, this makes the unwarranted assumption that the *dpp^FO^* and G-TRACE are identically sensitive to FLPase. In my experience, different flpout constructs show different sensitivities. The authors also seem to be implying that these are disc to disc differences in cell "lineage", but I doubt greatly whether stripe cells ever give rise to far anterior cells. Rather, the variation likely to be due to slight differences in Gal4 or FLPase expression, or simply a level of randomness in the excision events. In my hands, G-TRACE from Bloomington and *ptc*-Gal4 can even give rise to expression in the far posterior compartment, cells that have certainly not descended from anterior compartment stripe cells.

I cannot think of a way around this problem, short of building a new excision allele with a marker in the excised DNA. I am open to counter-arguments, but without something I cannot accept the authors' interpretation.

2) One difficulty is that Akiyama show that their *dpp*-gal4 technique removes most or all stripe Dpp from the dorsal wing pouch and hinge, and also greatly reduces pMad there, as early as 72 hours AEL (their Figure 3). Nonetheless, the dorsal pouch reaches a pretty normal-looking size by late third (although they did not measure pouch size alone, so it is possible there was a slight defect). If this is correct, then loss of the gradient and stripe do not affect growth from 72 hours on.

Either the authors need to disprove this, or they have to incorporate it into their discussion. Does the Akiyama allele version of the experiment lead to loss of Dpp and the pMad gradient in parts of the disc at 72 hours, and is growth in those regions affected or unaffected?

If Akiyama is correct, this should be mentioned. One possible explanation is that the authors might investigate is that the early pMad loss was not enough to increase brinker expression at early time points, as Akiyama only examined brinker at late third. Since the authors have Akiyama's allele, could they look? My thinking here is that the different results might not be due to whether stripe Dpp is lost, per se, but how much residual Dpp signaling is left from Dpp elsewhere in the disc, and whether that residual signaling is enough to suppress brinker expression during the growth phase.

3) In the Gal80ts experiment, the authors also need to show a control wing that is homozygous for the *dpp^FO^* allele, but reared continuously at 18*°*C.

Reviewer 2:

1) Figure 2 is a single addition to the GAL4 drivers explored in Extended Data 6 by Akiyama & Gibson, 2015; Figure 3 is an extension of Extended Data 4e of Akiyama & Gibson, 2015; Figure 4 is an extension of Extended Data 7 and lacks the temporal resolution for the Gal80ts experiment as well. There are no experiments to independently substantiate their claim of requirement of central stripe of Dpp for growth.

2) Usage of mCherry: NLS as a probe to study the activity of GAL4 domains: It has been demonstrated that GFP matures in about 5-20 mins, while mCherry matures in about 40-80 mins at 37°C. Slow maturation of mCherry is a property used in design of fluorescent timers (Khmelinskii et al., 2012). Maturation of mcherry is further slower at lower temperatures (~20-25°C). The kinetics of the marker (production/degradation) is crucial for marking the boundaries of GAL4s. Therefore, another probe should be used to confidently determine the boundaries of dpp-GAL4/ *ptc*-GAL4 domain. Ideally as Dpp is a secretory protein, the domain comparison should be studied with respect to Dpp mRNA and not protein.

3) Evans et al., 2009 using GTRACE show that the *ptc*-GAL4 marks the entire anterior compartment while *dpp*-GAL4 only marks a portion of anterior compartment although both show a similar real-time expression profile along the A/P boundary. It is important to "quantify" the variation observed in GTRACE to confidently negate the following possibility: Is *dpp* from the set of cells excluded by *dpp*-GAL4 lineage but included in *ptc*-GAL4 lineage (and *ci* GAL4) in the anterior compartment important for growth?

Reviewer 3:

The requirement of Dpp signaling for growth of the *Drosophila* wing has been a subject of much debate recently. Despite a large amount of evidence showing that Dpp is required for wing growth, and that it acts by repressing the growth-repressor Brinker, recent work has suggested that Dpp is only required for growth early in wing development, and not later on (as of mid-third instar). The manuscript here by Matsuda and Affolter revisits this issue. They show that Dpp is indeed required for wing disc growth, providing a technical explanation for the lack of a growth effect seen in Akiyama and Gibson 2015. In terms of temporal requirements, they show that Dpp is required as of early L3 for disc growth (although this was not really debated). In terms of spatial requirement, they find that discs where Dpp was removed only from the medial expression stripe display growth defects, indicating that the medial expression domain of Dpp is needed to support wing growth. Overall, the data quality are good. The manuscript analyzes the issue less in depth than the two other manuscripts that were co-submitted.

From Akiyama and Gibson, there is little debate whether Dpp is needed for growth at early 3rd instar. The debate is whether it is needed later on, as of 96h AEL (mid 3rd instar) (see Figure 2 of Akiyama and Gibson, where the reduction in size at 72h is by 50% and statistically significant, whereas the reduction in size at 96h is not significant). Hence Figure 4 presented here is not very novel, but instead should be repeated at 96h AEL.

---

## [Author Response]

Essential revisions:

Reviewer 1:

1) Since Akiyama already showed that removal from the entire anterior compartment reduces growth, the claim in the present study rests entirely on whether ptc-gal4 drives the excision of the conditional dpp allele in the stripe, or whether it drives excision more widely in the disc. This is a real worry, because the endogenous ptc gene is expressed at low levels throughout the anterior compartment. And as the authors (and others previously) show, ptc-gal4 can drive excision of a G-TRACE maker throughout the anterior as well.

Unfortunately, there is no direct way of telling where the dpp allele has been recombined. The conditional allele contains no marker of excision, and loss of dpp itself cannot be detected except where dpp expression is very strong, as Akiyama did.

The authors argue that there are some discs where ptc-gal4-driven G-TRACE is not excised throughout the anterior, and that therefore these discs must also be those where the conditional dpp allele is not recombined throughout the anterior. However, this makes the unwarranted assumption that the dpp^FO^ and G-TRACE are identically sensitive to FLPase. In my experience, different flpout constructs show different sensitivities. The authors also seem to be implying that these are disc to disc differences in cell "lineage", but I doubt greatly whether stripe cells ever give rise to far anterior cells. Rather, the variation likely to be due to slight differences in Gal4 or FLPase expression, or simply a level of randomness in the excision events. In my hands, G-TRACE from Bloomington and ptc-Gal4 can even give rise to expression in the far posterior compartment, cells that have certainly not descended from anterior compartment stripe cells.

I cannot think of a way around this problem, short of building a new excision allele with a marker in the excised DNA. I am open to counter-arguments, but without something I cannot accept the authors' interpretation.

We agree with the reviewer that our setup (and also Akiyama and Gibson’s setup) is based on the unwarranted assumption that the *dpp^FO^* allele and G-TRACE are identically sensitive to FLPase, and that random lineage of *ptc*-gal4-expressing cells (Figure 3 in old version, Figure 2 in new version) outside the stripe raises a serious worry on whether *dpp* is removed only from the stripe.

Considering the randomness of the *ptc*-gal4 lineages, we performed temporal *ptc*-Gal4 lineage experiments from second or from third instar larval stages using Gal80ts. We found that in both cases, the *ptc*-Gal4 lineage was strictly confined to a stripe of cells abutting the anterior-posterior compartment boundary, and that wing disc growth was severely affected (Figure 3). Thus, we focus on the role of *dpp* on wing disc growth from second or from third instar larval stages using such temporally staged larvae.

Nevertheless, and as the reviewer suggested, the ideal allele to directly monitor where *dpp* is removed would be a *dpp* excision allele with a marker in the excised DNA. We found such an allele (*dpp^FO-GFP^*) originally generated as an intermediate allele to generate the final *dpp^FO^* allele (Akiyama and Gibson, 2015). *dpp^FO-GFP^* contains the same FRT cassette as *dpp^FO^* and a ubiquitously expressed GFP construct (ubi-GFP) within the cassette (Figure 3—figure supplement 2). Thus, the regions in which *dpp* is excised will lack the GFP signal. Using this setup, we found that FLP/FRT mediated excision in the *dpp* locus was indeed restricted to the anterior stripe from second and from third instar larval stages (Figure 3—figure supplement 2). This result also strongly suggests that the temporal lineages of *ptc*-Gal4 indeed reflect the actual region where *dpp* is removed in this setup (Figure 3).

“Rather, the variation likely to be due to slight differences in Gal4 or FLPase expression, or simply a level of randomness in the excision events.”

We agree that the variation may be due to slight differences in Gal4 or FLPase expression, or simply a level of randomness in the excision events between different wing discs. Based on the temporal lineage experiments (Figure 3) (Figure 3—figure supplement 2), the variation is clearly derived from activation earlier than second instar larval stages. We now discuss these possibilities in our manuscript and the new temporal lineage results we obtained (Figure 3) (Figure 3—figure supplement 2).

2) One difficulty is that Akiyama show that their dpp-gal4 technique removes most or all stripe Dpp from the dorsal wing pouch and hinge, and also greatly reduces pMad there, as early as 72 hours AEL (their Figure 3). Nonetheless, the dorsal pouch reaches a pretty normal-looking size by late third (although they did not measure pouch size alone, so it is possible there was a slight defect). If this is correct, then loss of the gradient and stripe do not affect growth from 72 hours on.

Either the authors need to disprove this, or they have to incorporate it into their discussion. Does the Akiyama allele version of the experiment lead to loss of Dpp and the pMad gradient in parts of the disc at 72 hours, and is growth in those regions affected or unaffected?

If Akiyama is correct, this should be mentioned. One possible explanation is that the authors might investigate is that the early pMad loss was not enough to increase brinker expression at early time points, as Akiyama only examined brinker at late third. Since the authors have Akiyama's allele, could they look? My thinking here is that the different results might not be due to whether stripe Dpp is lost, per se, but how much residual Dpp signaling is left from Dpp elsewhere in the disc, and whether that residual signaling is enough to suppress brinker expression during the growth phase.

We agree that by removing *dpp* using *dpp*-gal4, Dpp protein and pMad signal was greatly reduced from the dorsal compartment during third instar larval stages, but that dorsal wing disc pouch still grew with minor growth defects (Akiyama and Gibson, 2015) (Figure 2 in this study). We incorporated this point in the discussion to explain the different growth phenotype caused by using either *dpp*-Gal4 or *ptc*-Gal4.

To ask why the dorsal compartment can still grow when using *dpp*-Gal4 despite reduced pMad and Dpp levels there, we analyzed Brk expression. We found that Brk was repressed in the ventral compartment where *dpp* was still present but slightly upregulated in the dorsal compartment where the majority of *dpp* was removed at the mid third instar larval stage (80hrAEL at 26°C) (Figure 3—figure supplement 4). Interestingly, the Brk upregulation in the dorsal compartment is not uniform but graded; it is lower in the ventral and higher in the dorsal part within the dorsal compartment (Figure 3—figure supplement 4). At the late third instar larval stage, the majority of *dpp* was eliminated and Brk was upregulated in both dorsal and ventral compartment, but again, Brk was not uniformly upregulated (Figure 3—figure supplement 4). Consistent with this finding, Omb has been shown to be weakly expressed in this setup (Akiyama and Gibson, 2015). These results suggest that the Dpp signal is not completely removed from the wing imaginal disc using *dpp*-Gal4. The graded Brk expression in the dorsal compartment is consistent with weak Dpp signal mediated by the ventral *dpp* stripe, and the lasting signal can explain the sustained growth there. In contrast, when *dpp* was removed by *ptc*-Gal4 during the third instar larval stages using Gal80ts, the majority of *dpp* was removed from the entire anterior stripe (Figure 3—figure supplement 3), and we indeed found severe growth defects (Figure 3).

Together, these results suggest that Akiyama and Gibson missed the critical role of the *dpp* stripe on wing disc growth due to the imprecise spatial removal of *dpp* using *dpp*-Gal4 during third instar larval stages. Based on the finding that there is a constant requirement of *dpp* for proper wing disc growth from the anterior compartment during third instar larval stages, Akiyama and Gibson proposed that a potential anterior *dpp* source outside the stripe was critical for wing disc growth (Akiyama and Gibson, 2015). However, the growth defects they observed are most likely due to the excision of *dpp* from the stripe.

3) In the Gal80ts experiment, the authors also need to show a control wing that is homozygous for the dpp^FO^ allele, but reared continuously at 18°C.

We performed the control experiment and found that a control wing (*dpp^FO^, ptc*-Gal4 / *dpp^FO^; tub*-Gal80ts / UAS-FLP) has no obvious phenotype at 17 °C (Figure 3—figure supplement 1), suggesting that Gal80ts effectively suppresses Gal4 activity at 17 °C.

Reviewer 2:

1) Figure 2 is a single addition to the GAL4 drivers explored in Extended Data 6 by Akiyama & Gibson, 2015; Figure 3 is an extension of Extended Data 4e of Akiyama & Gibson, 2015; Figure 4 is an extension of Extended Data 7 and lacks the temporal resolution for the Gal80ts experiment as well. There are no experiments to independently substantiate their claim of requirement of central stripe of Dpp for growth.

Although both *ptc*-Gal4 and *dpp*-gal4 are intensively used in the field and known to be expressed in a stripe of anterior cells, Akiyama and Gibson used only *dpp*-Gal4 to propose that the *dpp* stripe is not required for wing disc growth during third instar larval stages.

We followed up this question by using the same setup but a different diver line, namely *ptc*-Gal4. Contrary to their results, we found that the *dpp* stripe was indeed required for wing disc growth during third instar larval stages (Figure 3 in new version). Since we used an experimental setup as close as possible to the one used by Akiyama and Gibson, we can conclude that the different results are not due to different experimental setups but due to different driver lines (*dpp*-Gal4 vs *ptc*-Gal4).

“lacks the temporal resolution for the Gal80ts experiment”

Following the reviewer’s comment, we performed the temporal Gal80ts experiments to ask when *dpp* is eliminated from the stripe after a temperature shift. We found that, consistent with Akiyama and Gibson, the majority of *dpp* is removed from the entire stripe around 20 hr after temperature shift (Figure 3—figure supplement 3).

2) Usage of mCherry: NLS as a probe to study the activity of GAL4 domains: It has been demonstrated that GFP matures in about 5-20 mins, while mCherry matures in about 40-80 mins at 37°C. Slow maturation of mCherry is a property used in design of fluorescent timers (Khmelinskii et al., 2012). Maturation of mcherry is further slower at lower temperatures (~20-25°C). The kinetics of the marker (production/degradation) is crucial for marking the boundaries of GAL4s. Therefore, another probe should be used to confidently determine the boundaries of dpp-GAL4/ ptc-GAL4 domain.

To avoid maturation problem of a fluorescent protein, we compared mCherry fluorescent signal with anti-mCherry antibody staining (since maturation is dependent on oxidation within the FP structure and epitope recognition by antibody is independent on the mCherry maturation). We found that the mCherry fluorescent signal nicely overlaps with antibody staining in the early and late third instar larval stages (Figure 1—figure supplement 1).

Nevertheless, the critical setup to determine the boundaries of *ptc*-Gal4 relevant to our study would be to determine where FLP/FRT recombination occurs under the control of *ptc*-Gal4. As we discussed above, using the *dpp^FO-GFP^* allele, we also confirmed that *dpp* was indeed excised only from the anterior stripe when FLP/FRT recombination was induced in second or third instar larval stages (Figure 3—figure supplement 2).

Ideally as Dpp is a secretory protein, the domain comparison should be studied with respect to Dpp mRNA and not protein.

The Dpp antibody has been shown to recognize only Dpp prodomain but not mature ligands (Akiyama and Gibson, 2015). Thus, we used the Dpp prodomain antibody to compare the domains.

3) Evans et al., 2009 using GTRACE show that the ptc-GAL4 marks the entire anterior compartment while dpp-GAL4 only marks a portion of anterior compartment although both show a similar real-time expression profile along the A/P boundary. It is important to "quantify" the variation observed in GTRACE to confidently negate the following possibility: Is dpp from the set of cells excluded by dpp-GAL4 lineage but included in ptc-GAL4 lineage (and ci GAL4) in the anterior compartment important for growth?

As we discussed above, we found that the temporal *ptc*-gal4 lineages induced during second or third instar larval stages was strictly restricted to the anterior stripe of cells (Figure 3). Using the *dpp^FO-GFP^* allele, we also confirmed that *dpp* was indeed excised only from the anterior stripe when FLP/FRT recombination was induced in second or third instar larval stages (Figure 3—figure supplement 2). Thus, we focused on the role of the *dpp* stripe on growth from second or from third instar larval stages.

Reviewer 3:

The requirement of Dpp signaling for growth of the Drosophila wing has been a subject of much debate recently. Despite a large amount of evidence showing that Dpp is required for wing growth, and that it acts by repressing the growth-repressor Brinker, recent work has suggested that Dpp is only required for growth early in wing development, and not later on (as of mid-third instar). The manuscript here by Matsuda and Affolter revisits this issue. They show that Dpp is indeed required for wing disc growth, providing a technical explanation for the lack of a growth effect seen in Akiyama and Gibson 2015. In terms of temporal requirements, they show that Dpp is required as of early L3 for disc growth (although this was not really debated). In terms of spatial requirement, they find that discs where Dpp was removed only from the medial expression stripe display growth defects, indicating that the medial expression domain of Dpp is needed to support wing growth. Overall, the data quality are good. The manuscript analyzes the issue less in depth than the two other manuscripts that were co-submitted.

From Akiyama and Gibson, there is little debate whether Dpp is needed for growth at early 3rd instar. The debate is whether it is needed later on, as of 96h AEL (mid 3rd instar) (see Figure 2 of Akiyama and Gibson, where the reduction in size at 72h is by 50% and statistically significant, whereas the reduction in size at 96h is not significant). Hence Figure 4 presented here is not very novel, but instead should be repeated at 96h AEL.

In Figure 2, Akiyama and Gibson claim that the reduction in wing disc size is significant when removing *dpp* using hs-Flp at 96 AEL (Akiyama and Gibson, 2015). Together with the temporal *ci*-Gal4 experiments, they concluded that *dpp* is constantly required for wing disc growth during third instar larval. However, based on the results of the *dpp*-Gal4 experiments, they claimed that the *dpp* stripe is not required for wing disc growth during third instar larval stages, and propose a role for a potential anterior *dpp* source outside of the stripe region.

Since we used the same setup (*dpp^FO^*) as Akiyama and Gibson and temporal resolution (~20hr to remove *dpp*) is similar, we do not challenge the temporal requirement of *dpp* on growth during third instar larval stages. Thus, the debate here is whether the *dpp* stripe is indeed dispensable for wing disc growth, since Akiyama and Gibson used only *dpp*-gal4 to challenge the role of *dpp* stripe on wing disc growth. Furthermore, it remains unclear whether the *dpp* stripe is required for wing disc growth from these experiments. In our study, we used *ptc*-Gal4 to revisit the importance of the *dpp* stripe for growth. Using Gal80ts, we found that the majority of *dpp* was removed from the stripe around 20 hr after temperature shift (Figure 3—figure supplement 3), and that wing disc growth was severely affected at the late third instar larval stages (43hr later after temperature shift) (Figure 3). Since wing discs can grow for 24 hr until the majority of *dpp* is removed from the stripe (Figure 3—figure supplement 3), the growth defects that we observed likely reflect the absence of *dpp* from mid-third instar larval stage. Our result thus clearly demonstrates that the *dpp* stripe is indeed critical for wing disc growth during third instar larval stages.

Given the slow process of removing *dpp* by FLP/FRT mediated recombination compared to wing disc growth, it remains an open question whether the requirement of the *dpp* stripe on wing disc growth changes over time. It would be important in the future to acutely manipulate the endogenous morphogen gradient at the protein level to investigate the precise temporal requirement of *dpp* stripe. We are presently tempting such experiments using the previously published morphtrap approach as well as other newly designed synthetic receptors capable to trap Dpp fusion protein or Dpp protein itself.